# Physion++: Evaluating Physical Scene Understanding that Requires Online Inference of Different Physical Properties

**Hsiao-Yu Tung**[1,2*]   **Mingyu Ding**[1,3*]   **Zhenfang Chen**[4]   **Daniel M. Bear**[2]   **Chuang Gan**[4,5]
**Joshua B. Tenenbaum**[1]   **Daniel L. K. Yamins**[2]   **Judith Fan**[2]   **Kevin A. Smith**[1]
[1]MIT   [2]Stanford   [3]UC Berkeley   [4]MIT-IBM Watson AI Lab   [5]UMass Amherst
{sfish0101, kevin.smith3}@gmail.com   myding@berkeley.edu   jefan@stanford.edu

## Abstract

General physical scene understanding requires more than simply localizing and recognizing objects – it requires knowledge that objects can have different latent properties (e.g., mass or elasticity), and that those properties affect the outcome of physical events. While there has been great progress in physical and video prediction models in recent years, benchmarks to test their performance typically do not require an understanding that objects have individual physical properties, or at best test only those properties that are directly observable (e.g., size or color). This work proposes a novel dataset and benchmark, termed Physion++, that rigorously evaluates visual physical prediction in artificial systems under circumstances where those predictions rely on accurate estimates of the latent physical properties of objects in the scene. Specifically, we test scenarios where accurate prediction relies on estimates of properties such as mass, friction, elasticity, and deformability, and where the values of those properties can only be inferred by observing how objects move and interact with other objects or fluids. We evaluate the performance of a number of state-of-the-art prediction models that span a variety of levels of learning vs. built-in knowledge, and compare that performance to a set of human predictions. We find that models that have been trained using standard regimes and datasets do not spontaneously learn to make inferences about latent properties, but also that models that encode objectness and physical states tend to make better predictions. However, there is still a huge gap between all models and human performance, and all models' predictions correlate poorly with those made by humans, suggesting that no state-of-the-art model is learning to make physical predictions in a human-like way. These results show that current deep learning models that succeed in some settings nevertheless fail to achieve human-level physical prediction in other cases, especially those where latent property inference is required. Project page: https://dingmyu.github.io/physion_v2/

## 1   Introduction

The human visual system allows us to perceive rich contents in the dynamically changing physical world – from a static scene, we see objects, their shapes, sizes, colors, and textures. From objects' motions, we further see their mechanical properties, e.g., mass, friction, and elasticity, much more beyond simple object appearance. Knowing these mechanical properties enables us to adjust our prediction of the likely future – for instance, if we see a box sink into a couch cushion we might infer it is filled and so will be heavy when we pick it up. Any artificial system that is designed for general

---

*Equal contribution. This work was done when Mingyu was at MIT.

37th Conference on Neural Information Processing Systems (NeurIPS 2023) Track on Datasets and Benchmarks.

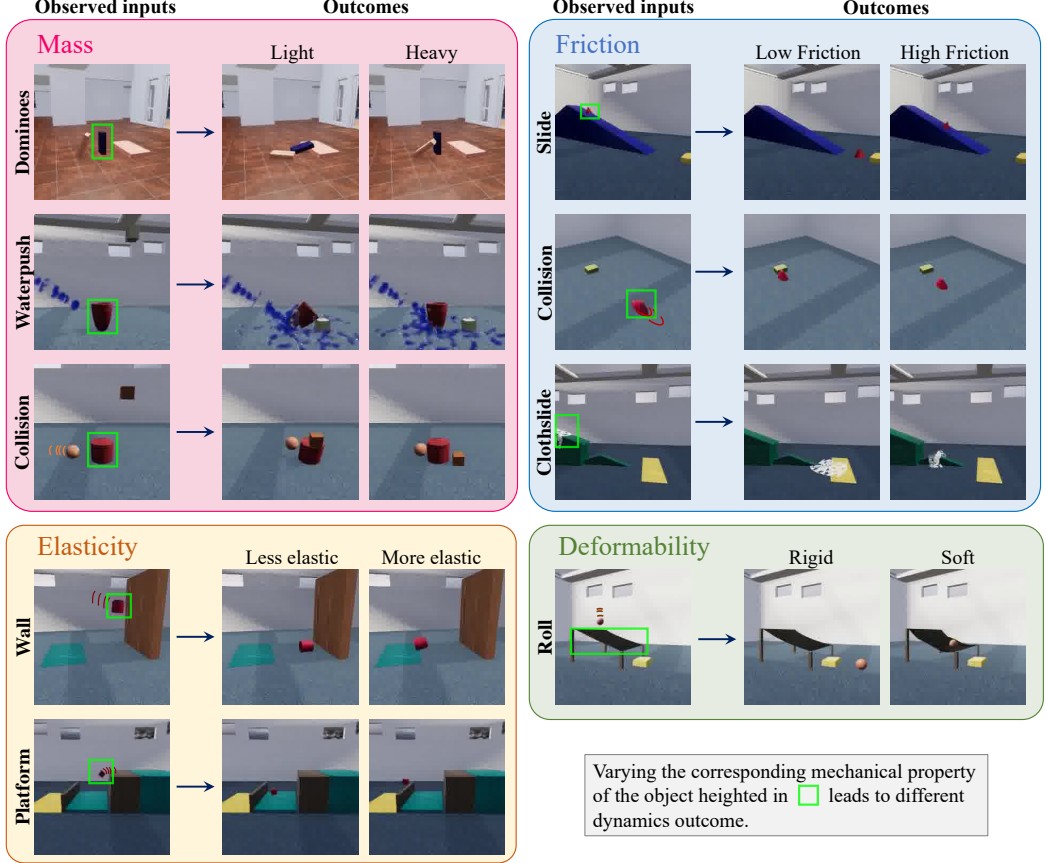

Figure 1: Different mechanical properties can lead to different physical outcomes. In this work, we design Physion++ consisting of 9 scenarios (Mass-Dominoes/Waterpush/Collision, Elasticity-Wall/Platform, Friction-Slide/Collision/Clothslide, and Deform-Roll) that try to probe visual physical prediction relevant to inferring four commonly seen mechanical properties – mass, friction, elasticity, and deformability. See Section 3.1 for more details.

physical prediction should also be able to infer the mechanical properties of objects and leverage that information for future predictions.

Prior benchmarks of physics [1, 2, 3, 4] have pushed the field to show that some models that purport to be good at physics are not actually capturing physically relevant outcomes, and that often model architectures that explicitly represent objects and their geometries outperform those that don't. However, such benchmarks do not require any understanding that objects have individuated properties, *e.g.*, all objects had the same density and elasticity. Agents (*e.g.*, dynamics models and human participants) in these benchmarks often make predictions based on visual appearance rather than physical properties, which can be ambiguous in real scenarios.

Take the mass-dominos task in Figure 1 (row1, left) as an example. The domino on the left (light brown) is falling down and might possibly hit the domino in the middle (dark blue), and the goal of the agent is to predict whether the dark blue domino would hit the light region on the right. However, the outcome of this scenario highly depends on the relative masses of the two dominos: (1) If the dark blue domino in the middle is much lighter, then as long as there is contact, then most likely it would fall down and hit the target region. (2) If the mass of the dark blue domino is much heavier, then it is possible that even if there is contact between the dominos, the dark blue domino would fail to fall down and therefore not hit the target region. The two scenarios have exactly the same shape and arrangement and thus would present the same observation without any information to disambiguate the masses. Therefore, how to evaluate agents' capability to predict latent mechanical properties of objects becomes a key issue.

To further evaluate visual physical prediction under circumstances where accurate physical prediction involves efficient inference of the latent mechanical properties of the objects, we introduce Physion++.

Table 1: Qualitative comparison between our Physion++ and other physics/dynamics-related video reasoning benchmarks. 'Diverse Phenomena' refers to whether the dataset covers diverse scenarios and types of properties, e.g., our dataset has 9 scenarios, including rigid bodies, fluids, soft bodies, objects of various shapes, and various physical properties. 'Few-shot Reasoning' means that our dataset allows physical properties to be judged from a few reference video frames, i.e., property inference.

| Dataset | Realistic | Object Dynamics | Physical Properties | Diverse Phenomena | Few-shot Reasoning |
|---|---|---|---|---|---|
| KITTI [5] | √ | × | × | × | × |
| Human3.6M [6] | √ | × | × | × | × |
| ShapeStacks [7] | √ | √ | √ | × | × |
| Cater [3] | × | × | × | √ | × |
| RoboNet [8] | √ | × | × | √ | √ |
| CLEVRER [4] | × | √ | × | × | × |
| CoPhy[9] | × | √ | √ | √ | × |
| PHYRE [10] | × | √ | × | √ | × |
| ESPRIT [10] | × | √ | × | √ | × |
| CRAFT [11] | × | √ | × | √ | × |
| Physion [1] | √ | √ | × | √ | × |
| IntPhys [2] | √ | √ | √ | × | × |
| Comphy [12] | × | √ | √ | × | √ |
| **Physion++ (ours)** | √ | √ | √ | √ | √ |

This dataset is built using the same engine and testing procedure as Physion [1], but is designed so that individual mechanical properties affect the physical outcomes. It consists of test suites on four commonly seen mechanical properties – mass, friction, elasticity, and deformability (see Figure 1), providing a more comprehensive challenge than previous benchmarks. For example, our dataset rigorously includes physical interactions between objects with different mechanical properties, *i.e.*, objects with varying masses, frictions, softness, and bounciness. The key difference compared to Physion [1] is thus that we further include extra frames (stimuli) depicting object interactions to enable agents to infer the mechanical properties before the actual prediction task. An example of our full observation is in Figure 2. We design the tasks such that, to accurately forecast the outcome in the prediction phase, the agents need to infer the mechanical properties in the inference phase. These additional mechanical properties also give us a chance to introduce more scenarios with soft objects and fluids (Figure 1) that are less explored in previous works like Physion.

Using Physion++, we test whether a set of state-of-the-art models that have been trained to make physical predictions also learn to make and leverage these property inferences. We further compare these models' outputs to human responses on each physical property to test if they perform physical predictions in a human-like way. **Our main contributions are thus three-fold:** (1) introducing Physion++, a novel benchmark to test mechanical property inference, (2) analyzing the performance of a set of state-of-the-art physical prediction models on this benchmark, and (3) comparing the performance of these models against human predictions on the same stimuli.

**Summary of key findings.** We found that all video models fail to capture human-level prediction across all four mechanical properties. Similar to the finding in [1], we found neural networks that encode objectness perform better than models that are not explicitly trained on these object-relevant tasks. We also find particle-based models which operate on 3D representations and have direct access to the latent mechanical properties perform better than models that operate on 2D representations. Yet, we found that no model was reliably using mechanical property inference, and therefore all models failed to correlate to human predictions. These results suggest that existing visual learning models, despite their success in a variety of video tasks, still fall short of human performance in general physics understanding tasks.

## 2 Related Work

**Physics and dynamics prediction benchmarks.** Humans can easily infer the rough dynamics and physical properties of objects from videos. For example, recent work [13, 14] has shown that people might approximate detailed 3D shapes with simple convex hulls or geometry for physics

understanding tasks. With the tremendous success of deep learning, many benchmarks [5, 6, 7, 3, 8, 4, 9, 10, 11, 1, 2, 12] are designed to evaluate the similar dynamics prediction capability of deep models. However, these benchmarks are either unrealistic, do not contain dynamics/physical properties, or only focus on a single scenario. The qualitative comparison between our Physion++ and other physics-related video reasoning benchmarks is shown in Table 1.

The next question is: How do we evaluate the physics knowledge learned by existing learning-based visual dynamics models and compare it to human behaviors on general physics understanding? Physion [1] provides a benchmark suite consisting of stimuli generated from realistic simulations on a wide range of dynamic scenes. The benchmark has helped evaluate progress towards human-like visual physical scene understanding algorithms and has provided insightful suggestions for model design. Yet, Physion includes physical prediction tasks that focus mainly on object shapes and arrangement, and not mechanical properties, while these properties can make a huge difference in the dynamics outcomes. Using Physion++, we specifically look at scenes that require making inferences about object properties from physics in order to get to the correct prediction – taking the best parts of Physion [1] and extending it to interrogate models of physics in deeper ways. Physion++ is purposefully designed to introduce the additional complexity that was ignored in Physion by incorporating different mechanical properties that influence the resulting physical interactions. This dataset includes a series of test suites focused on four fundamental mechanical properties: mass, friction, elasticity, and deformability. This broader scope presents a more comprehensive challenge compared to previous benchmarks, which is also different from what the Physion benchmark investigated.

**Mechanical property inference.** Recent work in material perception has demonstrated that people can quickly infer and use mechanical properties to make predictions about physical events [15, 16, 17]. While some work suggests that these properties are extracted using Bayesian inference [15], other work suggests that our visual system uses a set of task-specific heuristics to perform this estimation that could easily be learned by standard vision models. For example, the shape and motion fields of fluids could be used to estimate their viscosity [18, 19, 20]; deformation speed and contour of objects, as well as optical cues, could be used to estimate the elasticity of nonrigid objects when bending or pushing in [21, 22, 23]; and the trajectory length could be used to estimate the elasticity of an object bouncing around a room [14]. However, these visual heuristics leave the open question of how to choose which heuristics to use, and how to integrate them into a cohesive model that can account for human judgments in various physical reasoning tasks. There are also some works that integrate differentiable physics engines or learning-based physics simulators to perform property inference [24, 25, 26, 27, 28, 29] for video/physics property prediction and video question-answering tasks, but these models typically are based on a known dynamics model and do not learn physics from scratch.

**Video and dynamics prediction.** One popular hypothesis is that general intuitive physics knowledge can be learned through pixel prediction on large video datasets of natural physical scenes [30, 31, 32, 33, 34, 35, 36]. This provides a unified theory for intuitive physics knowledge learning and mechanical property inference without depending on task-specific heuristics [22, 23, 14], which is hard to obtain in general scenarios. This also provides a data-driven way to obtain the abstraction and approximation needed for a human-like intuitive physics engine [15], which is not limited by the availability of a realistic physics engine. These video models [37, 38, 39] are directly applicable to any video inputs, making it possible to handle natural dynamical scenes and even long-term video frame prediction on realistic scenes. Also, recent video diffusion models [39, 40] and transformer-based prediction models [37, 41, 42] show tremendous success.

Here we want to evaluate whether these models already encode general physics knowledge that includes mechanical property inference. We are also curious about how these models can explain human visual physical scene understanding, *i.e.*, if they perform physical predictions in a human-like way.

## 3 Benchmark Design

In this work, we introduce Physion++, which tests models' capabilities to infer four commonly seen mechanical properties from dynamics, including mass, friction, bounciness, and deformability. We first explain our stimuli design for each task, and then detail how we construct our benchmark.

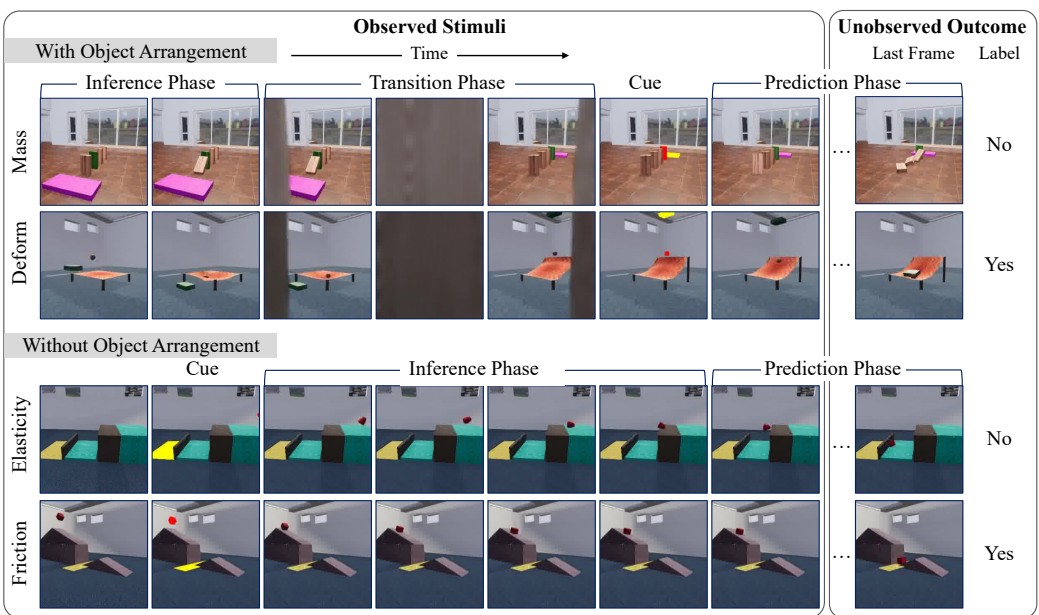

Figure 2: Stimuli are in the format of single videos. The videos consist of an inference period where artificial systems can identify objects' mechanical properties and a prediction period where the model needs to predict whether two specified objects will hit each other after the video ends.

## 3.1 Stimulus Design

We generate both the stimuli and the training data for visual dynamics learning algorithms with the ThreeDWorld simulator (TDW), a Unity3D-based environment [43]. We create 9 physical scenarios across 4 mechanical properties (see Figure 1), which are used to evaluate the model and human physics prediction ability that requires efficient inference of objects' mechanical properties from motion cues.

Stimuli are in the format of single videos, so they match better with circumstances that a real human might encounter in the real world. Following [1], we probe physics understanding with the **object contact prediction (OCP)** task, where the goal of the artificial systems is to predict whether two specified objects, cued in red and yellow in the middle of the video, will hit each other if physics continues to unfold after the video ends. In Table 1, we show Physion++ is more comprehensive than previous physical scene understanding benchmarks.

To provide sufficient information for artificial systems to infer objects' mechanical properties, we design the videos to consist of an inference period where the artificial systems can identify objects' mechanical properties and a prediction period where the model needs to predict whether two specified objects will hit each other after the video ends (see Figure 2). Before the prediction phase, we indicate the target two objects with blinking lights in red and yellow. And we task the model with the question "Will the red object hit the yellow object?"

The nine scenarios in the Physion++ dataset are (see Figure 1 for an example):

- 1.1 **Mass-Dominoes** – a sequence of collisions that depend on the arrangement and masses of objects may or may not cause the red block to fall on the yellow mat
- 1.2 **Mass-Waterpush** – a collision between liquid and a rigid body red object may or may not cause that object to collide with a yellow object
- 1.3 **Mass-Collision** – a collision between an object and the red object may or may not cause the red object to move out of the path of a falling yellow object
- 2.1 **Elasticity-Wall** – the red object is thrown towards a wall and may collide with the yellow mat, depending on their placement, trajectories, and objects' elasticities
- 2.2 **Elasticity-Platform** – the red object is thrown towards a surface and may bounce onto or miss the yellow mat

3.1 **Friction-Slide** – the red object slides down a ramp and may make contact with the yellow object, or may stop beforehand due to friction

3.2 **Friction-Collision** – the red object starts with velocity and slides across the floor and may make contact with the yellow object, or may stop beforehand due to friction

3.3 **Friction-Clothslide** – a piece of red cloth slides down a ramp and may land on the yellow mat, or may under- or over-shoot it

4.1 **Deform-Roll** – the red object is dropped onto a piece of cloth and may either sink in or roll off in order to possibly hit the yellow object

We aim to avoid strong associations between superficial visual cues with the final YES/NO outcome by designing "paired" trials, where the paired video scenes are visually identical in the first frame during the prediction phase yet they unfold into different event outcomes due to different latent physical properties assigned to the objects in the videos.

In some scenarios, the inference and prediction phases can be included in the same video (*e.g.*, Elasticity-Platform, Friction-Slide, and Friction-Clothslide). However, in many cases, the physical event that provides information in the inference phase irrevocably changes the configuration of objects so that there is no way to use the inferred information for future predictions (*e.g.*, judging mass from seeing one domino topple into another leaves them both on the floor; Figure 2, top row). In these cases, we include a "transition phase": a curtain slides in to block the scene, then while the scene is occluded the objects are rearranged for the prediction phase, and finally the curtain moves out of the way. The cueing of the two target objects is done immediately after the transition phase, followed by a short observation of the rearranged objects in motion.

Further details of stimulus creation are provided in the Supplemental Materials.

## 3.2 Dataset Design

Each scenario of Physion++ consists of three stimulus sets: dynamics training, readout fitting, and testing. We describe them below, and provide further details in the Supplemental Materials.

**Dynamics training set.** The training dataset is provided so the agents can learn the dynamics of the environment, and thus learn representations that can be discriminative enough to distinguish whether the red object will hit the yellow object in the testing dataset. We generate 2,000 trials for each mechanical property without YES/NO contact labels for dynamics pretraining. For each physical scenario, in half of the trials the red object contacts the yellow object, and in half there is no contact, so as to ensure the balance of learning. Scenarios with an object rearrangement phase were included in the dynamics training so that models could learn that it was unsurprising for objects to be rearranged behind the curtain, but that objects retained their properties.

**Readout fitting set.** The readout fitting set is a small dataset containing 192 trials (96 pairs) for each mechanical property, used to map the dynamic representation learned in the training set to YES/NO of the OCP task. Trials were balanced for equal contact / no contact events to avoid bias.

**Testing set.** The final testing benchmark also consists of 192 trials (96 pairs) for each mechanical property. It has the same visual and physical statistics as the readout fitting set so that the learned mapping from the readout set can be directly evaluated on the test set.

## 4 Experiments

### 4.1 Model Performance

**Prediction Models.** We evaluate state-of-the-art video models that fall into four main categories representing different levels of learning versus built-in knowledge. 1) Video models that learn a visual encoder and physics prediction model through pixel-wise prediction: **MCVD** [39]. 2) Visual encoders learned from supervised pretraining on ImageNet [44] that are designed to output responses using extended MLPs, which are trained in an unsupervised way on the benchmark readout data: **pRESNET-mlp** [45], **pVGG-mlp** [46], and **pDEIT-mlp** [38]. 3) Video models that learn physics prediction models on top of object-centric representations, obtained from self-supervised image pretraining on its encoder: **SlotFormer** [41] and **ALOE** [37] 4) 3D particle-based graph neural

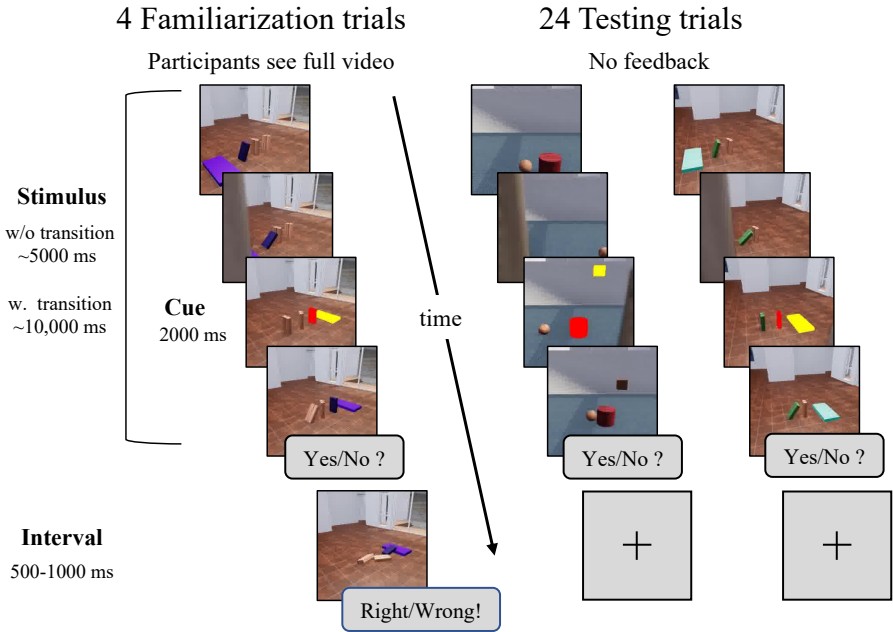

Figure 3: Demonstration of the human task. Trial structure for the familiarization trials (left) and test trials (right) indicating the Cue, Stimulus, and Inter-trial periods.

network that takes ground truth 3D state and latent properties of objects as inputs (without a visual encoder, i.e. assume perfect observability of physical dynamics) and learns object transformation and dynamics as explicit 3D flow: **DPI-Net** [47]. More details are provided in the Appendix.

**Training and readout protocols.** We use 2 different training protocols and 3 readout protocols. For training, we test both models that learn dynamics separately for each mechanical property ('separate'), and models that learn unified dynamics models for all scenarios at once ('unified').

We test the models on three variants of each video. The 'w/ property' video is the standard video described in the stimulus design section. The 'w/o property' video does not include the initial frames of the video where the critical object first interacts in the scene, and therefore there is no property inference possible. This setting was used to test whether the model performance was in fact driven by using mechanical properties, or based on predictions without that information. Finally, the 'fully observed' video extends the standard video to include the red object contacting or missing the yellow object. Similar to [1], we use these videos as a test of the quality of the representations learned by each model – can we read out the OCP answer even if it is directly observed? We trained a separate readout model for each video type.

**Training and Evaluation Pipelines.** For all models, we first pretrain the learning of dynamics using the dynamics training set by predicting future frames under a future prediction loss. Then on the readout fitting set, we perform rollout using the pretrained dynamics model and extract representations from both observed and predicted frames. Based on these representations we train a readout model as logistic regression. Finally, we evaluate the model by extracting representations and reading out the OCP task from the testing set.

**Results.** The results of baseline models on Physion++ are shown in Table 2. We can observe that: (1) Training on all mechanical properties (unified) typically leads to inferior performance to individual training (separate), suggesting that learning joint models of dynamics produces worse predictions than learning individual scene dynamics. (2) Most models perform only slightly above chance, suggesting that this task is challenging for all current models. (3) Most models with property inference perform similarly to those without property inference, indicating that these models are not utilizing physical property inference. (4) With full video observations, the prediction performance generally improves, suggesting the learned representations do contain additional object information that could be predicted by their dynamics models. (5) DPI-Net achieves the best overall performance as it leverages ground

Table 2: Results of model performance on the Physion++ dataset. For testing videos, 'w/' and 'w/o property' denote if the model makes predictions based on the property inference stage. 'fully observed' – model using the ground truth prediction frames.

| Subset | Training | Testing | MCVD | DeiT-mlp | ResNet-mlp | VGG-mlp | ALOE | SlotFormer | DPI-Net |
|---|---|---|---|---|---|---|---|---|---|
| Mass | unified | w/ property | 52.6 | 50.5 | 48.0 | 53.6 | 52.1 | 53.6 | 52.1 |
| | | | 54.2 | 56.3 | 54.7 | 55.2 | 53.1 | 58.9 | 75.9 |
| | separate | w/o property | 53.1 | 55.0 | 54.2 | 56.3 | 52.6 | 57.3 | 60.1 |
| | | fully observed | 67.1 | 69.8 | 67.7 | 60.4 | 56.3 | 66.7 | 96.9 |
| Friction | unified | w/ property | 52.6 | 54.2 | 50.5 | 50.0 | 52.1 | 50.5 | 53.8 |
| | | | 52.6 | 52.6 | 52.1 | 52.6 | 56.8 | 54.2 | 52.5 |
| | separate | w/o property | 54.2 | 52.6 | 52.1 | 51.6 | 53.1 | 54.2 | 54.7 |
| | | fully observed | 56.3 | 53.6 | 53.6 | 54.2 | 54.2 | 58.3 | 84.6 |
| Elasticity | unified | w/ property | 53.1 | 52.1 | 55.2 | 51.6 | 51.6 | 52.6 | 51.0 |
| | | | 52.6 | 55.2 | 54.2 | 52.6 | 52.6 | 58.3 | 52.1 |
| | separate | w/o property | 51.6 | 55.2 | 54.7 | 52.6 | 53.1 | 56.8 | 52.4 |
| | | fully observed | 55.7 | 59.4 | 54.7 | 54.2 | 53.1 | 64.6 | 95.8 |
| Deformability | unified | w/ property | 58.9 | 56.3 | 55.2 | 52.1 | 51.6 | 52.1 | 55.1 |
| | | | 57.8 | 57.3 | 59.4 | 61.5 | 51.0 | 55.2 | 60.2 |
| | separate | w/o property | 58.3 | 53.6 | 58.3 | 59.9 | 51.0 | 56.3 | 62.6 |
| | | fully observed | 60.4 | 72.9 | 68.8 | 69.8 | 58.3 | 70.8 | 91.3 |
| Overall | unified | w/ property | 54.3 | 53.3 | 52.2 | 51.8 | 51.9 | 52.2 | 53.0 |
| | | | 54.3 | 55.4 | 55.1 | 55.5 | 53.4 | 56.7 | 60.2 |
| | separate | w/o property | 54.3 | 54.1 | 54.8 | 55.1 | 52.5 | 56.2 | 57.5 |
| | | fully observed | 59.9 | 63.9 | 61.2 | 56.0 | 55.5 | 65.1 | 92.2 |

truth 3D information. VGG-mlp achieves good performance on the deformability set and SlotFormer achieves good performance on mass and elasticity sets.

In summary, most models fail to perform physical property inference, except DPI-Net in the mass scenario (60.1% for w/o property and 75.9% for w/ property), potentially indicating this model has learned to infer just the property of mass. This suggests that future work focusing on better physical property inference for deep models would be worth exploring in order to build systems that more generally reason about physical scenarios.

## 4.2   Human experiments

We additionally gathered predictions from human participants in order to test whether the models (a) reached human levels of performance, and (b) performed the task in a way similar to people, as measured by the same pattern of errors. 200 participants (50 for each of the mechanical properties) were recruited from Prolific and paid $15.50 per hour for their participation. We selected 192 trials from each scenario, then shuffle and split the 192 trials into 8 disjoint sets, where all sets have half of the trials containing contact between the red and yellow objects, and half do not. Each participant was shown 32 stimuli drawn from one of these disjoint sets of trials drawn from a single mechanical property, and we ensure participants do not observe both parts of any matched pair of trials. Data is balanced, so each participant will see 16 stimuli with the ground truth label 'YES' and 16 with 'NO'. These studies were conducted in accordance with the MIT and UC San Diego IRBs.

**Task procedure.** The structure of our task is shown in Figure 3. Each trial began with a fixation cross, which was shown for a randomly sampled time between 500ms and 1500ms. The stimuli will start playing right after the fixation. To indicate which of the objects shown was the agent and patient object, participants were shown the agent and patient objects were overlaid in red and yellow respectively, at the first frame during the prediction phase of the video for 2000ms. During this time, the overlay flashed on and off with a frequency of 2Hz. After this, the stimulus will keep playing until it ends and gets removed, and the response buttons were enabled. Participants proceeded to the next trial after they made a prediction by selecting either 'YES' (the red and yellow objects would touch) or 'NO' (they would not). The order of the buttons was randomized between participants. Before the main task, participants were familiarized with 4 trials that were presented similarly to the

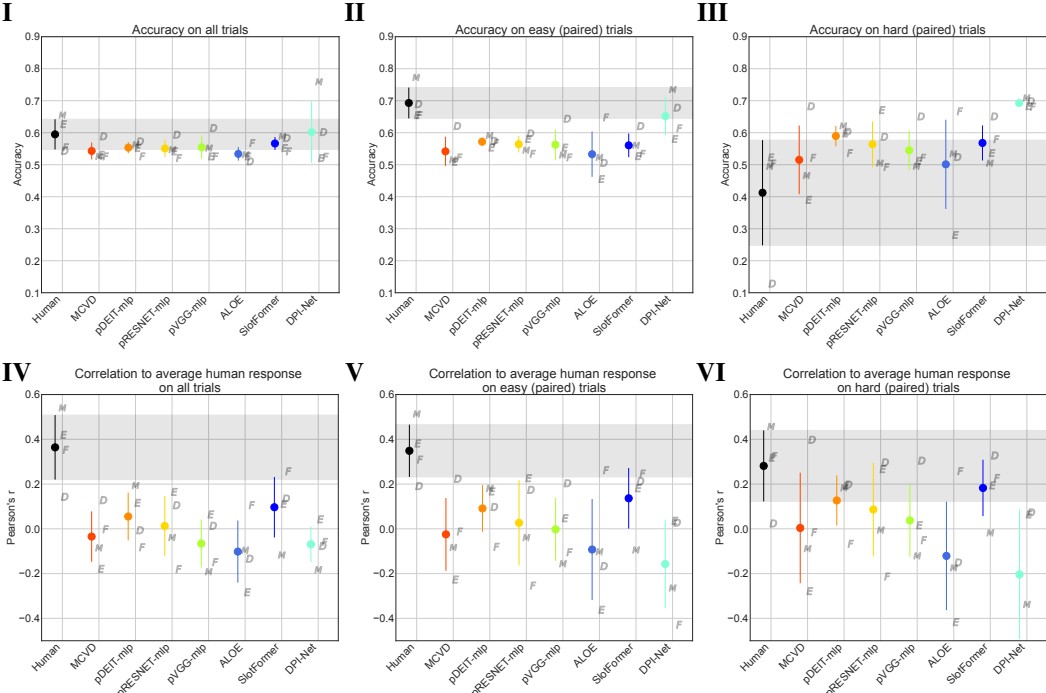

Figure 4: Comparisons between humans and models. First row: task accuracy (I, II, III), Second row: Pearson correlation between model output and average human response (IV, V, VI). We evaluate the metric in three settings: with the whole testing dataset (all trials), with trials (in pairs) that humans perform particularly well (above 67% accuracy; easy trials) and poorly (below 33% accuracy; hard trials). We use the letters 'D, F, M, E' to denote the task deformability, friction, mass, and elasticity, respectively.

test trials, except the full stimulus movie and accuracy feedback were presented after participants indicated their prediction. Familiarization trials were always presented in the same order. After the test trials were completed, basic demographics were collected from participants. Finally, participants were informed of their overall accuracy.

**Results and human-model comparisons.** Following [1], we compare a model's outputs under the 'separate' training and 'with property' testing protocols to human responses on each physics property's testing stimuli with two standard metrics: overall accuracy and Pearson correlation between model and average human responses across stimuli. See Figure 4 I and IV, respectively. From the results, we found that humans are performing reliably above chance but not particularly well at the task (acc: 60%), yet all video models are performing slightly worse than humans. Only DPI-Net, which operates on ground truth states from the physics engine performs comparably to humans. While this is a challenging task, using trials with a range of human performance (including many below chance) allows us to test whether model and human predictions are based on the same competencies, or whether they excel on different trials.

We found that no model correlated well with humans. SlotFormer holds the highest correlation with human predictions (r=0.12 vs. split half human correlation r=0.37), suggesting that object-centric approaches may be better suited for physics inference than alternatives, but they are not sufficient for producing human-like predictions.

**Performance on easy/hard trials.** We further examine the model and human performance on trials where humans perform particularly well and particularly poorly. This allows us to understand whether the models are differentiating trials that humans find trivial from those they find difficult, or whether model performance is driven by features not used by people. We evaluate on the "easy" trials by selecting stimuli from some of the same-initial-configuration pairs (see Stimulus Design for more details) if one of the stimuli in the pair has mean human performance higher than 66.7%, and we also evaluate the "hard" trials where one of the paired stimuli has mean human performance lower than 33.3%. These results are shown in Figure 4 II and V for the easy trials, and III, VI for the hard trials.

Table 3: Ablation study on the number of training trials per property.

| # Trials | DeiT-mlp | ResNet-mlp | ALOE | SlotFormer |
|----------|----------|------------|------|------------|
| 200 | 49.0 | 50.0 | 50.5 | 51.6 |
| 500 | 53.6 | 51.6 | 52.1 | 54.2 |
| 1000 | 54.7 | 54.7 | 54.2 | 56.3 |
| 2000 | 55.4 | 55.1 | 53.4 | 56.7 |

From these results, we see that all models appear to be making predictions in a way different from people: they all underperform humans on the easy trials, but outperform humans on the difficult trials. These results suggest that existing visual learning models still fail to capture human behaviors in general physics understanding tasks, especially in cases where efficient inference of the underlying physics properties is required.

**Performance on each of the four physical properties.** According to Figure 4, most models perform better than humans in the deformability category, while their performances for all the other three categories are worse than human judgment. This might result from the fact that humans are better at learning intuitive physics hence the deformability property is harder to predict than the other three (see hard trials in Figure 4 III). However, these models do not utilize physical property inference and learn from data distributions, therefore the preferences for the four categories are less pronounced.

**Performance on different number of training trials per property.** We conducted additional ablative experiments with different numbers of trials by selecting four representative models (DeiT-mlp, ResNet-mlp, ALOE, and SlotFormer). The models learn dynamics separately for each mechanical property with property inference, and we report the average performance of the four mechanical properties. The results are shown in Table 3. We can see that when the data size is small (200), the performance of all models is close to random guessing. As the number of training data increases, the accuracy of the all models improve and gradually saturate, with only minor performance increases from 1000 to 2000 trials. Therefore, we believe that 2000 training trials for each property are reasonable, and that any performance differences we observe are related to architectural differences. In the future, we will continue to enrich our data to further investigate asymptotic performance across these and other models.

## 5 Discussion

Physical scene understanding itself is a challenging task, requiring knowledge of visual perception, spatial relation, physical property understanding and inference, distance and dynamics estimation, etc. Despite recent successes from neural-network-based video frame prediction models, none of these models correlate with human predictions on Physion++. This indicates that current computational models do not make online inferences about physical properties. This could be for a number of reasons. It could be that property inference is a difficult task; these models were not designed with this inference in mind, and 2,000 scenarios might not be enough for this capability to emerge. In addition, these models might struggle with video prediction that involves multi-stage long-term dependencies. Our current findings suggest that while the accuracy levels may be both close to 0.6, the correlation between human and model performances is low, indicating their proficiency in distinct domains. Humans and models excel at different scenarios and trials. It shows that although the models outperform humans in some scenarios, they still have huge room for improvement, e.g., they still fail in many scenarios that humans can easily handle.

The videos in our tasks often involve several phases: inference events, the transition phase, and the test phase, which causes the videos to last for 13-20 seconds. On the other hand, these models are often trained and tested on shorter videos. We believe with future video and dynamics model development that enables accurate predictions on multi-phase videos, our proposed Physion++ dataset can again serve as the benchmark to evaluate how well these models can be used to infer mechanical properties and how well their performance correlate with humans. The proposed dataset has no ethical or societal issues on its own, except those inherited from physical scene understanding.

**Acknowledgement.** This work was supported by ONR MURI grant N00014-22-1-2740.

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

# A   Project Page and Dataset Release

We have released the Physion++ dataset at our project page. All three subsets (training set [2], readout fitting set [3], and testing set [4]) and the human data [5] are publicly available. We also post a list of all test filenames in the test set (192 entries per physical property). The dataset is organized as follows.

```
# Download and unzip the three data files
-- Training_data
   ...
-- Readout_data
   ...
-- Testing_data
   -- Folder organized by physical properties  # folders named with copy0 and copy1
    indicate two matched trials (e.g., the same video location from copy0 will have a
    different property/outcome in copy1 but the same initial conditions)
       -- Subfolder organized by scenarios and objects  # e.g. bouncy_platform-
    use_blocker_with_hole=0-target=cube
          -- [id].json  # object id and instance information over time
          -- [id].pkl  # meta information of the physical event and the video
          -- [id]_image.mp4  # raw input, RGB video
          -- [id]_seg.mp4  # segmentation masks of the video
          -- [id]_map.png  # indicating the yellow and red objects
   # Below are lists of test filenames and ids (192 entries per physical property)
   -- Mass.csv
   -- Friction.csv
   -- Elasticity.csv
   -- Deformability.csv
```

Listing 1: Organization of the dataset.

The .json file contains the object id and segmentation mask of each instance over time. The .pkl file contains the meta information of the scene, including 1) the dynamic friction, static friction, initial position, initial rotation, mass, elasticity, color, and mesh of each object; 2) the camera matrix and projection matrix for each frame; 3) angular velocities, positions, rotations, and velocities of all objects over time; 4) the collision events, including the object ids, relative velocities, time and states; 5) the trial seed used to generate the video, the label for the video, and 'start_frame_for_prediction', which is a timestamp indicating the part of the video before the timestamp is visible, and the part after the timestamp is required to be predicted.

**Author statement.** We confirm that we bear all responsibility in case of any violation of rights during the collection of the data or other work, and will take appropriate action when needed, e.g. to remove data with such issues.

**Hosting, licensing, and maintenance plan.** We host the dataset on Amazon AWS. We ensure access to the data and will provide the necessary maintenance. All products created as part of this project is shared under the MIT license. We used a number of third-party software packages, each of which typically has its own licensing provisions. Only TDW [43] was used in the creation of the dataset; all others were models used for assessment. Table 4 contains a list of these licenses for many of the packages used.

# B   Scenario Details

**Mass-Dominoes.** This scenario starts with an inference phase, where a set of "dominoes" (equally sized cuboids standing long end up) are placed approximately in a row with semi-random spacing and orientations. One of the dominoes is visually marked with a different texture, indicating a different material. At the start of the video, a domino at one end starts to fall as if it had been pushed over, starting a sequence of the dominoes being pushed over. The video continues past the point where the textured domino is hit by or hits one of the other dominoes, providing information about the textured dominoes' mass. A transition phase is then required (as all relevant objects have toppled)

---

[2]Training data: https://physion-v2.s3.amazonaws.com/train_data.zip
[3]Readout data: https://physion-v2.s3.amazonaws.com/readout_data.zip
[4]Testing data: https://physion-v2.s3.amazonaws.com/test_data.zip
[5]Human data: https://physion-v2.s3.amazonaws.com/human_data.zip

Table 4: Table of open-source code used.

| Name | URL | License |
|------|-----|---------|
| MCVD [39] | https://github.com/voletiv/mcvd-pytorch | MIT License |
| ALOE [37] | deepmind/object_attention_for_reasoning | Apache License 2.0 |
| ResNet [45] | https://github.com/pytorch/vision | BSD 3-Clause License |
| DeiT [38] | https://github.com/facebookresearch/deit | Apache License 2.0 |
| VGGNet [46] | https://github.com/pytorch/vision | BSD 3-Clause License |
| DPI-Net [47] | https://github.com/YunzhuLi/DPI-Net | N/A |
| TDW [43] | https://github.com/threedworld-mit/tdw | BSD 2-Clause License |
| SlotFormer [41] | https://github.com/pairlab/SlotFormer | MIT License |

and the dominoes are reset. In the reset scene, there is a mat on the floor, and one of the dominoes is indicated as the 'red' object while the mat is the 'yellow' object. The textured domino is placed in the chain so that its mass will influence the chain of dominoes: e.g., if it is too heavy it would not topple when another domino strikes it, but if not it will continue the sequence of collisions and cause the red domino to land on the yellow mat.

**Mass-Waterpush.** The inference phase begins with an object at rest and a stream of water shooting towards the object as if out of a hose. This stream may move the object, and thus give information about its mass. The transition phase then occurs where the object is moved to another location (and placed upright if it has tipped over) and marked as 'red'. In addition, a new object is added falling from midair, marked as 'yellow'. The yellow object might be positioned above the red object, in which case, depending on the masses, the water might knock the red object out of the way or fail to move it. Or the yellow object might be positioned further along the path defined by the water stream, so that the stream might cause the red object to slide into the yellow object, or fail to move.

**Mass-Collision.** The scenario is identical to the Mass-Waterpush scenario, except that instead of a stream of water that pushes the object, a ball rolls into and collides with the object in both the inference and testing phases.

**Elasticity-Wall.** In the inference phase, an object is in ballistic motion towards a wall, bounces off, and lands on the floor. This provides information about the elasticity of the collision. The transition phase then occurs, and the object is again placed in ballistic motion towards the wall (which does not move) and marked as the 'red' object, with a 'yellow' mat being placed on the floor. Depending on the elasticity, after bouncing off of the wall, the red object may land on, or over- or under-shoot the yellow mat.

**Elasticity-Platform.** This scenario contains a raised platform that ends and drops onto a surface, with a wall at the end of that surface, followed by a mat (marked as 'yellow') on the floor. It starts with an object (marked as 'red') bouncing onto the platform and continuing to bounce/slide to its edge. This provides information about the objects' elasticity, and thus how it will bounce when it falls onto the surface. The key question is whether the red object will touch the mat. In some cases, the wall is short, so the object must have high elasticity to bounce over and hit the mat. In other cases, the wall is tall but has a slot at the bottom; in these cases if the object bounces too much it will hit the wall and stop, but if it is less elastic it will slide through the slot and contact the mat. In this way we decorrelate the elasticity from the outcome.

**Friction-Slide.** There were two subtypes of scenarios for Friction-Slide. In both cases, an object (marked as 'red') is positioned near the top of a ramp and begins to slide down. In the 'gap' situation, there is a divot in the slope, and the 'yellow' mat is positioned in that gap; thus depending on the friction of the red object it might slide into the gap and contact the mat or fly over the gap and miss it. In the 'no-gap' situation, there is no divot, and the 'yellow' object is placed in the runout area of the slope; thus the red object might stop before hitting the yellow object or might continue to slide into it. We use both subtypes so that there is not a correlation between low- or high-friction objects and the outcome. No transition phase is needed in this case; the inference can be performed from the first part of the video when the object is sliding down the slope.

**Friction-Collision.** This scenario starts with the key 'red' object sliding along the floor, providing information about the friction of that object. The transition phase occurs and then the red object is reset to a different position with a new velocity, and a 'yellow' object is dropped from above (similar

to the Mass-Collision and Mass-Waterpush scenarios). The crucial judgment is whether the friction of the red object will slow it down just enough so the yellow object will land on it, or whether the red object will under- or over-shoot the mark.

**Friction-Clothslide.** This scenario is identical to the Friction-Slide scenario, except instead of a rigid object sliding down the ramp, the 'red' object is a piece of cloth.

**Deform-Roll.** This scenario starts with a cloth hanging from posts either vertically or horizontally, and a key object is launched or dropped on the cloth respectively. This provides information about the deformability of the cloth. The transition phase occurs and the cloth is now hung from posts at an angle. The critical 'red' object is dropped from above the cloth, and the 'yellow' object is set in one of two places. It is either positioned on the ground towards the base of the cloth, so that it is important to determine whether the red object will sink into the cloth, or roll off of it and hit the yellow object. Or the yellow object is dropped from above the red object, so that if the cloth is deformable enough both will sink in and touch, but if it is not, the red object will roll off before the yellow and they will never contact.

The examples of all 9 scenarios are shown in our project page.

## C   Details of the Dataset

In some scenarios, the inference and prediction phases can be included in the same video (*e.g.*, Elasticity-Platform, Friction-Slide, and Friction-ClothSlide). However, in many cases, the physical event that provides information in the inference phase irrevocably changes the configuration of objects so that there is no way to use the inferred information for future predictions (*e.g.*, judging mass from seeing one domino topple into another leaves them both on the floor at the top row of Figure 2 in the main paper). In these cases, we include a "transition phase": a curtain slides in to block the scene, then while the scene is occluded the objects are rearranged for the prediction phase, and finally the curtain moves out of the way. And the cueing of the two target objects is done immediately after the transition phase, followed by a short observation of the rearranged objects in motion. Examples of all 9 scenarios are shown in our project page.

**Training set.** The training dataset is used for the agents to learn dynamics prediction, and the learned representations can be discriminative enough to distinguish whether the red object hit the yellow object, and can generalize to the testing dataset. We generate 2000 trials for each mechanical property without YES/NO labels for dynamics pretraining. For each physical scenario, we make half of the trials where the output answers are YES and half of them are NO, so as to ensure the balance of learning.

**Readout fitting set.** The readout fitting set is a small dataset containing 192 trials used to map the dynamic representation learned in the training set to YES/NO of the video question-answering (*i.e.*, OCP) task.

**Testing set.** The final testing benchmark consists of 192 trials (96 pairs) for each mechanical property. We aim to avoid strong associations between superficial visual cues with the final YES/NO outcome by designing the readout and test dataset to be "paired" trials, where the paired video scenes are visually identical in the first frame during the prediction phase yet they unfold into different event outcomes due to different latent physical properties assigned to the objects in the videos. We achieve this by fixing object configuration during the prediction phase and regenerating the stimuli with uniformly sampled mechanical values on a target object until we get one stimulus with a positive outcome, and the other with a negative outcome. For each scene configuration, we sample a maximum of 5 different property values, and we drop scenes where we sample all true or all false outcomes. It has the same overall visual and physical statistics as the readout fitting set so that the learned mapping from the readout set can be directly evaluated on the test set.

## D   Pipelines and settings

For each video, we truncate (or pad) both the inference phase and the prediction phase to 160 frames, and sub-sample the videos by a factor of 5 for training the representation or dynamics models. All frames are resized to $128 \times 128$ to reduce the computational cost. For SlotFormer [41] and ALOE [37], we first pre-train the object-centric models STEVE [48] and MONet [49] on all

Table 5: Ablation study on implicit physical property inference.

| Method | DeiT-mlp | ResNet-mlp | ALOE | SlotFormer |
|---|---|---|---|---|
| implicit (OCP only) | 55.4 | 55.1 | 53.4 | 56.7 |
| explicit as auxiliary | 54.2 | 55.7 | 54.7 | 56.7 |

sub-sampled frames for scene decomposition, and extract all slot representations for subsequent training. The dynamics models are then trained on the slot representations from the training set under future prediction loss. For MCVD, the frames are directly fed into the model for dynamics prediction under image reconstruction loss. For pRESNET-mlp, pVGG-mlp, and pDEIT-mlp, we leverage pretrained ResNet50 [45], VGG16 [46], and DeiT-small [38] on ImageNet as our feature extractors. For DPI-Net, we represent the scene with particle representation provided by the annotation. For the oracle model with property inference, we add the ground-truth property values into the attribute embedding input of DPI-Net in both training and testing. For the model without property inference, we simply mask all property values with padding zero vectors. For the full video observed, we feed the ground-truth particle-based representation to the model. We calculate the distance $d_{min}$ between the closet particles in the two target objects. We consider the two objects will contact if $d_{min}$ is smaller than a threshold $\eta$ that is learned from the training set. $\eta$ is set to 0.075. For other parameters, we follow the same setting as Physion [1].

For models to generate YES/NO responses from their learned representations, we use the readout fitting set for the models to learn to map from their latent representation to the target response. We perform rollout to generate future scene representations (e.g. feature maps for image-based methods, or object slots for object-centric methods) based on the inference phase in the readout set. We implement a multilayer perceptron (MLP) with intermediate dimensions of 256 and 64 as our readout model, which is trained on rollout scene representations from the readout set to classify whether the two cued objects contact. All experiments were run on 8 NVIDIA TITAN X GPUs using the Adam optimizer and a learning rate of 1e-4. The models learned on the training and readout sets are then evaluated on our final benchmark (testing set) by applying the learned visual representations and the readout model. The best testing results among all readout training epochs are reported.

## E    Explicit and implicit physical property inference

We selected four representative models (DeiT-mlp, ResNet-mlp, ALOE, and SlotFormer) for the experiment. A parallel layer of MLP is added at the end of the model (two-branch multitask: OCP and explicit property estimation) during the readout fitting process. With explicit property estimation as an auxiliary task, we report the performance comparison with our original setting as in Table 5. With explicit inference, performance barely improves, suggesting that the networks do not have access to the properties even when prompted, not that they understand properties but fail to use them for prediction.

## F    Datasheets for dataset

Here are our responses in reference to the Datasheets for Datasets [50] standards.

**Motivation.**

- **For what purpose was the dataset created?** To measure deep models' physical future prediction abilities and latent property inference capabilities, and compare these to predictions made by humans.

- **Who created the dataset and on behalf of which entity?** The authors listed on this paper, including researchers from MIT, Stanford, UC Berkeley, MIT-IBM Watson AI Lab, and UMass Amherst.

- **Who funded the creation of the dataset?** The various granting agencies supporting the above-named researchers, including both grants to the PIs as well as individual fellowships for graduate students and postdoctoral fellows involved with the project.

**Composition.**

- **What do the instances that comprise the dataset represent?** Each instance is a video of a simulated physical scene (e.g. a tower of blocks as it either collapses or remains steady), together with some metadata about that video, including map-structured metadata with segmentation maps and information about object-object collisions at each timepoint.

- **How many instances are there in total?** The dynamics prediction model training dataset consists of 2000 examples for each of the 4 physical properties. The OCP readout fitting dataset consists of 192 examples per each of the 4 physical properties. The test dataset (on which human responses were obtained) consists of 192 examples per physical property.

- **Does the dataset contain all possible instances or is it a sample of instances from a larger set?** Data is generated by a simulator; in a sense, the set of datapoints we created is an infinitesimally small subset of data that *could* have been generated. However, we are all here releasing all the examples we did actually generate.

- **What data does each instance consist of?** It consists of a video depicting a physical situation (e.g a tower of blocks falling over), together with simulator-generated metadata about the situation.

- **Is there a label or target associated with each instance?** For the training dataset, there are no labels. For both the OCP readout fitting dataset and the human testing dataset, there are binary labels describing whether the red object collided with the yellow zone during the duration of the trajectory.

- **Is any information missing from individual instances?** No.

- **Are relationships between individual instances made explicit?** Yes. All data is provided in a simple data structure that indicates which instances of data are connected with which instances of metadata.

- **Are there recommended data splits?** Yes, for each of the scenarios in the datasets, there are three splits: (a) a large training split for training physical prediction models from scratch; (b) a smaller readout-training set that is to be used for training the yes/no binary readout training as described in the paper, and (c) the test dataset on which human responses were obtained.

- **Are there any errors, sources of noise, or redundancies in the dataset?** We have not found any as of this publication. As these are discovered, they will be fixed and versioned.

- **Is the dataset self-contained, or does it link to or otherwise rely on external resources?** It is self-contained.

- **Does the dataset contain data that might be considered confidential?** No.

- **Does the dataset contain data that, if viewed directly, might be offensive, insulting, threatening, or might otherwise cause anxiety?** No.

- **Does the dataset relate to people?** No.

**Collection Process.**

- **How was the data associated with each instance acquired? What mechanisms or procedures were used to collect the data? How was it verified?** Videos (for training, readout fitting, and human testing) were generated using the TDW simulation environment. Online crowdsourcing was used to obtain human judgements for each testing video. During the creation of the simulated videos, the researchers looked at the generated videos by eye to verify if the scenarios were correct (e.g. actually depicted the situations desired by our experimental design). Prior to running the actual data collection procedure for humans, we verified that the experimental websites were correct by having several of the researchers complete the experiment themselves.

- **Who was involved in the data collection process and how were they compensated?** PIs, students, and postdocs generated simulator-generated videos. For human responses, 200 participants (50 for each of the mechanical properties) were recruited from Prolific and paid $15.50 per hour for their participation.

- **Over what timeframe was the data collected?** All simulator-generated scenarios were created and human data was collected during the second half of 2022.
- **Were any ethical review processes conducted?** All human data collection was approved by UC San Diego IRB.

**Preprocessing, clearning and labelling.**

- **Was any preprocessing/cleaning/labeling of the data done?** We reviewed the test scenarios to make sure we videos with non-informative situations were not included (e.g., one of the key objects is fully occluded during the entirety of the video). No other preprocessing was done, and labeling was produced automatically by the system.

**Uses.**

- **Has the dataset been used for any tasks already?** Yes, the participants in the human experiments used the data for the single purpose for which it was designed: obtaining detailed characterization of human judgments about physical prediction and latent property inference in simple scenes.
- **Is there a repository that links to any or all papers or systems that use the dataset?**. No other papers use the dataset yet.
- **What (other) tasks could the dataset be used for?** None.
- **Is there anything about the composition of the dataset or the way it was collected and preprocessed/cleaned/labeled that might impact future uses?** No.
- **Are there tasks for which the dataset should not be used?** The dataset can only be used to measure abilities of humans or models to make physical prediction based on latent property inference.

**Distribution.**

- **Will the dataset be distributed to third parties outside of the entity (e.g., company, institution, organization) on behalf of which the dataset was created?** Yes it will be completely publicly available via our project page and the links listed thereupon.
- **How will the dataset will be distributed?** It will be available via links to the project page, and which will refer to permanent Amazon S3 resources.
- **When will the dataset be distributed?** Immediately.
- **Will the dataset be distributed under a copyright or other intellectual property (IP) license, and/or under applicable terms of use (ToU)?** The dataset and associated code will be licensed under the MIT license.
- **Have any third parties imposed IP-based or other restrictions on the data associated with the instances?** No.
- **Do any export controls or other regulatory restrictions apply to the dataset or to individual instances?** No.

**Maintenance.**

- **Who is supporting/hosting/maintaining the dataset?** The dataset is hosted on Amazon S3 resource. The associated Amazon S3 account is the institutional account for the CogTools lab (at Stanford).
- **How can the owner/curator/manager of the dataset be contacted?** The corresponding author of the paper can be contacted via email as described in the front page of the paper.
- **Is there an erratum?** No. If needed, any future errata will be posted on the project page.
- **Will the dataset be updated (e.g., to correct labeling errors, add new instances, delete instances)?** As this dataset becomes used by a larger audience, we will review the instances for errors that users uncover. These errors will be corrected as they are discovered on an ongoing basis.

- **Will older versions of the dataset continue to be supported/hosted/maintained?** If newer versions of the dataset are created, these will only be in additional to the existing data. Old versions will be maintained indefinitely.

- **If others want to extend/augment/build on/contribute to the dataset, is there a mechanism for them to do so?** No. Making contributions to this dataset requires a very detailed understanding of a variety of components and how they interconnect – physics simulators, scenario generation modules, online psychophysical experimentation platforms, etc. – and we do not contemplate allowing third parties to (e.g.) add new examples of physical scenarios.

**Structured metadata.** We have not created structured metadata for our project in a format like that in schema.org or DCAT as yet, because we expect that through the review feedback process, the exact structure of what metadata we should provide may change. We will be happy to do this once review is complete. In the meantime, all of our data is available through our project page, which provides a certain level of metadata about the project that we think is appropriate for the review process.

**Dataset identifier.** At the moment, we provide access to the dataset via Amazon S3 links that are visible via our project page. We have not yet pushed out data into a standard data repository or created a DOI for it. This is because we expect the specifics of how the data is made available to develop during the paper review process. Once this is complete, we will push the data into a standardized data repository and generate a DOI for it.

