# OpenReview forum: "Physion++: Evaluating Physical Scene Understanding that Requires Online Inference of Different Physical Properties"
_NeurIPS.cc/2023/Track/Datasets_and_Benchmarks — NeurIPS 2023 Datasets and Benchmarks Poster_

### Official Review · Reviewer_vxoX · 2023-07-22
**Paper Review**

**Rating:** 6
**Confidence:** 2
**Correctness:** I'm worried about the human performan…
**Clarity:** Yes.

**Strengths:**

- The problem is intuitive and interesting. It tries to address the common problem to learn intuitive physics. And it can be impactful.
- The experiments are extensive. It includes many baselines and human study.

**Additional Feedback:**

N/A

**Documentation:**

Yes.

**Ethics:**

No.

**Limitations:**

It does not look like the paper has discussed any limitations explicitly.

**Opportunities For Improvement:**

Honestly I think the paper is very good. However, my primary concern is why the human performance is so low in section 4.2. I'm happy to raise my rating if the rebuttal can address my concern.

- Based on my understanding, all questions and YES/NO question, so the random performance is table 4 and figure 2 is always 0.5, right?
- In this case, why the human performance is so low on figure 4? On all trails, the accuracy is 0.6, which is marginally above 0.5. It becomes worse for hard trails -- the human performance comes to 0.4, which is below random performance. Can you explain why?

If this is true, it does not make sense to me to construct such a dataset. Humans cannot solve these intuitive physics task. Why should we expect machines to solve it?

> (L306-307) none of these models match the human-level prediction ability on Physion++

I think this conclusion is also invalid in this case. On Physion++, both models and humans are close to random performance. The room for models to improve is quite limited.

**Relation To Prior Work:**

Yes.

**Summary And Contributions:**

This paper proposes Physion++, which tries to do visual physical prediction based on different mechanical properties -- mass, friction, elasticity and deformability. It consists of 9 different scenarios. It is by far the most complete physics/dynamics-related video reasoning benchmark. Both model performance and human studies are conducted in the experiments section.

---

> ### Author Response · Authors · 2023-08-23
> **Rebuttal by Authors (Part 1/2)**
>
> Dear reviewer vxoX,
>
> We would like to begin by expressing our sincere gratitude for your thorough review of our paper. We greatly appreciate your suggestions, which are crucial in improving the quality of our paper. The questions you raised are insightful, which we believe have been carefully clarified and addressed as follows.
>
> **Q1. [However, my primary concern is why human performance is so low in section 4.2. I'm happy to raise my rating if the rebuttal can address my concern.]** \
> Good question! We appreciate your consideration of increasing your score and we will provide a detailed explanation below.
>
> > Based on my understanding, all questions and YES/NO question, so the random performance in table 2 and figure 4 is always 0.5, right? In this case, why the human performance is so low on figure 4? On all trails, the accuracy is 0.6, which is marginally above 0.5. It becomes worse for hard trails -- the human performance comes to 0.4, which is below random performance.
>
> Yes, the random performance is 0.5 for Table 2 and Figure 4(I). However, for Figure 4(II) and (III), the human performance can be much lower and higher, as the (easy or hard) evaluation data is defined by the mean human performance (L289-L293 in our main paper). We evaluate the "easy" trials by selecting stimuli from some of the same-initial-configuration pairs if one of the stimuli in the pair has mean human performance higher than 66.7%, and we also evaluate the "hard" trials where one of the paired stimuli has mean human performance lower than 33.3%. Therefore, hard trials are trials in which humans have a high probability of failure, with a human performance of less than 50%.
>
>
> > If this is true, it does not make sense to me to construct such a dataset. Humans cannot solve this intuitive physics task. Why should we expect machines to solve it?
>
> - Physical scene understanding itself is a challenging task, requiring knowledge of visual perception, spatial relation, physical property understanding and inference, distance and dynamics estimation, etc. It's non-trivial for the model design with property inference and is also difficult for humans.
> - Our current findings suggest that while the accuracy levels may be both close to 0.6, the correlation between human and model performances is low, indicating their proficiency in distinct domains. Humans and models excel at different scenarios and trials. It shows that although the models outperform humans in some scenarios, they still have huge room for improvement, e.g., they still fail in many scenarios that humans can easily handle.
> - We have also conducted a more thorough analysis of the data for highlighting instances where either humans or DPI-Net outperform the other. The representative videos are uploaded to our [webpage](https://dingmyu.github.io/physion_v2/). We can see that humans tend to have trouble dealing with unusual camera angles, contrasting physical properties (e.g., two objects with one very heavy and one very light, or very large elasticity), and long-term predictions. But DPI-Net can handle these scenarios better, especially for trials in mass and deformability. Instead models may fail in simple situations (due to lack of distance perception and intuitive physics) that are easier for humans to understand.
>
> We will incorporate a detailed breakdown of the trials or properties in which humans exhibit better performance compared to baseline models, as well as cases where each model demonstrates superiority over human performance. We believe that this analysis will provide valuable insights into the capabilities and limitations of different methods and contribute to a more comprehensive physical scene understanding.

---

> > ### Author Response · Authors · 2023-08-23
> > **Rebuttal by Authors (Part 2/2)**
> >
> > **Q2. ["(L306-307) none of these models match the human-level prediction ability on Physion++". I think this conclusion is also invalid in this case. On Physion++, both models and humans are close to random performance. The room for models to improve is quite limited.]** \
> > Sorry for the confusion. In Figure 4(I), we observe DPI-Net has an overall comparable or even better performance with the human-level prediction. We believe that 60% prediction accuracy is not bad, because the task in Physion++ is really challenging. Physical property inference is very difficult for both human participants and models, especially since we set up paired data (see Lines 172-175) to interfere with model selection. And we believe that after enough training, humans can further improve their prediction accuracy, and the models also have huge room for improvement.
> >
> > Another key information is that, as we responded to Q1, our current findings suggest that while the accuracy levels of human and DPI-Net may be close to 0.6, the correlation between human and model performances is low, indicating their proficiency in distinct domains. Humans and models excel at different scenarios and trials. It shows that although the models outperform humans in some scenarios, they still have huge room for improvement, e.g., they still fail in many scenarios that humans can easily handle.
> >
> > We will carefully revise the above clarifications in our revision, and discuss in detail the scenarios that different models and humans are better at handling. We will gather more human feedback and further enhance the predictive models in the future, and we believe our dataset can serve as a general benchmark for the complex and challenging physical property inference task.
> >
> > \
> > We wish that our response has addressed your concerns, and turns your assessment to the positive side. If you have any questions, please feel free to let us know during the rebuttal window. We appreciate your suggestions and comments! Thank you!
> >
> > Best, \
> > Authors

---

> > > ### Comment · Reviewer_vxoX · 2023-08-26
> > >
> > > Thanks for the rebuttal! It addresses my concern. I appreciate it and I'll raise my rating to 6.

---

> > > > ### Author Response · Authors · 2023-08-29
> > > > **Sincerely thank you for your support and recognition of our paper!**
> > > >
> > > > Dear reviewer,
> > > >
> > > > Thanks for raising the rating. We deeply appreciate the extensive time and effort you invested in reviewing our paper. We are glad to see that our response has addressed your concerns. We will carefully revise our paper according to your suggestions, which have significantly enhanced the quality and clarity of our work!
> > > >
> > > > Please don’t hesitate to let us know if there are any additional clarifications we can offer.
> > > >
> > > > Best, \
> > > > Authors

---

### Official Review · Reviewer_WLS8 · 2023-07-24
**synthetic dataset for future state prediction of physical object with visual and non visual dataset**

**Rating:** 6
**Confidence:** 3
**Correctness:** The paper seems to be correct.
**Clarity:** The write-up could be improved. What …

**Strengths:**

- Paper addresses a relevant problem of using physical properties to predict state of a scene.
- Human evaluations are provided

**Additional Feedback:**

comments provided above.

**Documentation:**

url, dataset details, were are provided in the webstite.

**Ethics:**

There is no ethics concern.

**Limitations:**

- The model prediction and the human prediction accuracies are only a little above the random threshold. This might point to the task being ill-defined by the dataset. It was better if a few (2-5) were asked to predict the state of the same scene and their agreement could be analyzed. For an ill-defined dataset, they would not agree most of the time.

- Its unclear if predictions from a 3d simulated model for this task will generalize to the real world. It depends on the quality of the renders and the physics simulation of the 3d engine.

- Its an incremental improvement over existing benchmark Physion.

**Opportunities For Improvement:**

- The paper can be better written. It was not clear what the input and expected outputs of the models trained on the dataset would be.
- The model prediction and the human prediction accuracies are only a little above the random threshold. This might point to the task being ill-defined by the dataset. It was better if a few (2-5) humans were asked to predict the state of the same scene and their agreement could be analyzed. For an ill-defined dataset, they would not agree most of the time.


**Relation To Prior Work:**

There is a proper comparison present with the previous works.

**Summary And Contributions:**

The paper presents a benchmark called Physion++ which builds on a published benchmark Physion. The improvement of Physion++ from Physion is that it adds physical properties like mass, friction, elasticity, and deformability. As input model is given the properties of the objects, and sequence of rendered image frames, and can also include 3d geometry information and the goal is to predict some "state of the scene". The paper compares human prediction accuracies with various models.

---

> ### Author Response · Authors · 2023-08-23
> **Rebuttal by Authors (Part 1/2)**
>
> Dear reviewer WLS8,
>
> Thank you for the positive comments and insightful suggestions! Your insightful questions and valuable suggestions have been immensely helpful in enhancing our paper's quality.
>
> **Q1. [The paper can be better written. It was not clear what the input and expected outputs of the models trained on the dataset would be.]** \
> The training details could be found in Section D (Pipelines and settings) in our Appendix. For each video, we truncate (or pad) both the inference phase and the prediction phase to 160 frames, and sub-sample the videos by a factor of 5 for training the representation or dynamics models. All frames are resized to $128 \times 128$ to reduce the computational cost.
>
> For models to generate YES/NO responses from their learned representations, we use the readout fitting set for the models to learn to map from their latent representation to the target response. We perform rollout to generate future scene representations (e.g. feature maps for image-based methods, or object slots for object-centric methods) based on the inference phase in the readout set. We implement a multilayer perceptron (MLP) with intermediate dimensions of 256 and 64 as our readout model, which is trained on rollout scene representations from the readout set to classify whether the two cued objects contact. Please refer to section D (Pipelines and settings) in our Appendix for more details about each model. In our revision, we will provide more comprehensive details of the input data and the corresponding expected outputs for the models trained on our dataset.
>
> **Q2. [The model prediction and the human prediction accuracies are only a little above the random threshold. This might point to the task being ill-defined by the dataset. It was better if a few (2-5) humans were asked to predict the state of the same scene and their agreement could be analyzed. For an ill-defined dataset, they would not agree most of the time.]** \
> Good suggestion! We believe our dataset is not ill-defined and we explain it in the following aspects:
> - Physical scene understanding with property inference itself is a challenging task, requiring knowledge of visual perception, spatial relation, physical property understanding and inference, distance and dynamics estimation, etc. It's non-trivial for the model design with property inference and is also difficult for humans.
> - Our current findings suggest that while the accuracy levels may be both close to 0.6, the correlation between human and model performances is low, indicating their proficiency in distinct domains. Humans and models excel at different scenarios and trials. It shows that although the models outperform humans in some scenarios, they still have huge room for improvement, e.g., they still fail in many scenarios that humans can easily handle.
> - We have also conducted a more thorough analysis of the data for highlighting instances where either humans or DPI-Net outperform the other. The representative videos are uploaded to our [webpage](https://dingmyu.github.io/physion_v2/). We can see that humans tend to have trouble dealing with unusual camera angles, contrasting physical properties, and long-term predictions. But DPI-Net can handle these scenarios better, especially for trials in mass and deformability. Instead models may fail in simple situations (due to lack of distance perception and intuitive physics) that are easier for humans to understand.
> - We have also conducted human testing as you suggested and observed that human prediction of physical properties is basically consistent. We appreciate your suggestions and are conducting human agreement analysis across all physical properties and scenarios. We will post all quantitative results here as soon as they are available, thank you!

---

> > ### Author Response · Authors · 2023-08-23
> > **Rebuttal by Authors (Part 2/2)**
> >
> > **Q3. [Its unclear if predictions from a 3d simulated model for this task will generalize to the real world. It depends on the quality of the renders and the physics simulation of the 3d engine.]** \
> > Good question! The challenge of generalizing predictions from a 3D simulated model to real-world scenarios is indeed a crucial consideration, and we appreciate your insight into this matter.
> > - It's true that the issue of generalization is a significant concern for any synthetic dataset. While we acknowledge the inherent limitations in replicating real-world dynamics, we have made earnest efforts to maximize the representational quality of our dataset within the confines of a simulation environment.
> > - We understand that the quality of renders and the accuracy of physics simulations within the 3D engine are pivotal factors influencing the potential for real-world generalization. While achieving a perfect one-to-one correspondence between simulation and reality is challenging, we have implemented rigorous procedures to ensure the fidelity of our simulations, striving for a balance between accuracy and computational feasibility.
> > - While we cannot completely replicate real-world dynamics, we believe that by providing a comprehensive and diverse dataset with varying mechanical properties and interactions, we are facilitating the development and evaluation of models that can demonstrate robustness and adaptability across a spectrum of scenarios. Our aim is to contribute to the advancement of AI models that, while not entirely immune to the synthetic-to-real gap, can bridge this gap to a meaningful extent.
> >
> > We will be dedicated to refining our dataset and methods in the future to enhance the generalizability to real world of our work.
> >
> >
> > **Q4. [Its an incremental improvement over existing benchmark Physion.]** \
> > Thanks. We respectfully push back the comment that Physion++ is an incremental improvement over Physion.
> >
> > Physion++ is purposefully designed to introduce the additional complexity that was ignored in Physion by incorporating different mechanical properties that influence the resulting physical interactions. This dataset includes a series of test suites focused on four fundamental mechanical properties: mass, friction, elasticity, and deformability (illustrated in Figure 1). This broader scope presents a more comprehensive challenge compared to previous benchmarks, which is also entirely different from what the Physion benchmark investigated.
> >
> > The key is physical property inference. Instead of Physion with no physical property inference, we test scenarios where accurate prediction relies on estimates of properties such as mass, friction, elasticity, and deformability, and where the values of those properties can only be inferred by observing how objects move and interact with other objects or fluids.
> >
> > Moreover, our dataset contains 9 different scenarios, and 4 physical properties, and includes objects of various shapes, rigid bodies, soft bodies, and fluids, which Physion does not have.
> >
> >
> > \
> > Thanks again for your time and effort! For any other questions, please feel free to let us know during the rebuttal window.
> >
> > Best, \
> > Authors

---

### Official Review · Reviewer_zV2y · 2023-07-26
**A promising novel benchmark with some limitations**

**Rating:** 7
**Confidence:** 4

**Strengths:**

The strengths of the paper are listed below: \
**Importance** : As machine learning systems proliferate more to support cognition, decision making and autonomous action in the real world, and seek to seamlessly integrate in human society, they are expected to behave in a common-sensical, human-like manner including being able to deduce about mechanical properties of objects from their dynamics, sensed and perceived visually, and using them to make physical predictions. The Physion++ benchmark proposed in this paper is a step in this direction. \
**Significance**: I expect the dataset and the benchmark to be useful to the research community working in this area (as well as of interest to the broader research community), challenging existing models and approaches as well as spurring more research in this important direction. \
**Methodology**: In general, the methodology is ok. (see below) \
**Clarity**: The paper is largely written well (see below). \
**Reproducibility**: Should be reproducible given the information in the paper, supplementary and the website (see below).


**Additional Feedback:**

I'm happy to discuss further with the authors on improving the work. In general, I like the direction of this work.

**Clarity:**

Yes, the paper is well written.


**Correctness:**

I have some misgivings about the benchmark and the experiment design which I discuss above.

**Documentation:**

The paper, the supplementary, and the website together have enough detail to support reproducibility, though I couldn't find any supporting code.

**Ethics:**

None.


**Limitations:**

I don’t think there are any direct negative societal implications. Other limitations and opportunities for improvement are mentioned in the response to the previous question.


**Opportunities For Improvement:**

### Major
**Appropriateness/ Solvability** : For a novel benchmark to aid in spurring new research, it needs to be proper: while being difficult for the current SOTA, it should not be _unsolvable_. The human performance on the binary task is close to 0.6 (random = 0.5) - dropping below random (0.4) for hard trials (defined by human performance) and increasing to 0.7 on easy trials. While the visual perception and cognition tasks don't seem to be difficult for humans, the prediction is. _This raises a fundamental question as to whether there is enough information to make the prediction at all or not?_ In this sense, I feel the benchmark should’ve been designed better. \
**Missing Baselines** : The OCP task can be made conditionally independent of the visual information given the location, size, and tracks of the objects involved, detection of the contact event and the latent mechanical properties (even if specified qualitatively or comparatively in a paired setting). It seems that the difficulty of inferring the latter (observability?) and using them for reasoning (including inherent uncertainties involved in extrapolating on motion, force and dynamics) are what makes this task challenging both for humans and the machines. I’d have preferred baseline models and human experiments that decoupled these two aspects of the problem. (I’m happy to elaborate but want to keep the review short). \
__Human Explanations, Theories at Play__ : I also recommend that the authors think about collecting explanations from the human subjects which will allow them to gather data on what theories humans bring into play by themselves, and see if suggestions/ prompting/ guidance to use the right theory helps improve performance. This can be very helpful in rationalizing expectations from and the designing of such machine systems.  \

### Minor

**Clarity**: In Figure 4, bottom row paper (IV, V, VI): Kindly explain the human performance  correlation with average human performance. \
**Reproducibility**: Sharing the code to reproduce the results and analysis in the paper will go a long way in helping the community quickly and easily reproducing the work. Strongly urged. \


**Relation To Prior Work:**

Relation to prior art, the novelty and contributions are adequately discussed.


**Summary And Contributions:**

The paper introduces a novel benchmark called Physion++ to evaluate physical prediction models, i.e. models that seek to solve video understanding tasks involving mechanical physics needing the inference of and utilization of mechanical properties of involved objects in the videos. The proposed benchmark purports to be more realistics, diverse, involving dynamics between objects influenced by their differing material properties in a few-shot reasoning framework. SOTA models are evaluated on the benchmark as also human performance. Results and analysis is shared showing that in spite of tremendous progress, extant SOTA models don’t perform well on the benchmark falling significantly short of human performance on the corresponding tasks on the benchmark. The benchmark is expected to spur research in the creation of models capable of better physical scene understanding.

---

> ### Comment · Reviewer_zV2y · 2023-08-20
> **Finalized rating with a strong concern**
>
> I don't see a rebuttal from the authors. I have also gone through the other reviews. While I'm leaving the rating as is for now (no reason to change), I do have some strong concerns (appropriateness of the tasks in the benchmark, and missing 'reasonable' baselines). I'm happy to have a conversation about this with other reviewers and the AC, or leave it to the judgment of the AC.

---

> > ### Author Response · Authors · 2023-08-23
> > **Rebuttal by Authors (Part 1/2)**
> >
> > Dear reviewer zV2y,
> >
> > Thanks for the active discussion and positive comments! We apologize for being late due to additional experiments as well as data/human studies. Your insightful questions and valuable suggestions have been immensely helpful in enhancing our paper's quality.
> >
> > **Q1. [Appropriateness/ Solvability : For a novel benchmark to aid in spurring new research, it needs to be proper: while being difficult for the current SOTA, it should not be unsolvable. The human performance on the binary task is close to 0.6 (random = 0.5) - dropping below random (0.4) for hard trials (defined by human performance) and increasing to 0.7 on easy trials. While the visual perception and cognition tasks don't seem to be difficult for humans, the prediction is. This raises a fundamental question as to whether there is enough information to make the prediction at all or not? In this sense, I feel the benchmark should’ve been designed better.]** \
> > Thanks for your question! We answer your questions from the following aspects.
> > - Physical scene understanding itself is a challenging task, requiring knowledge of visual perception, spatial relation, physical property understanding and inference, distance and dynamics estimation, etc. It's non-trivial for the model design with property inference and is also difficult for humans.
> > - Our current findings suggest that while the accuracy levels may be both close to 0.6, the correlation between human and model performances is low, indicating their proficiency in distinct domains. Humans and models excel at different scenarios and trials. It shows that although the models outperform humans in some scenarios, they still have huge room for improvement, e.g., they still fail in many scenarios that humans can easily handle.
> > - We have also conducted a more thorough analysis of the data for highlighting instances where either humans or DPI-Net outperform the other. The representative videos are uploaded to our [webpage](https://dingmyu.github.io/physion_v2/). We can see that humans tend to have trouble dealing with unusual camera angles, contrasting physical properties (e.g., two objects with one very heavy and one very light, or very large elasticity), and long-term predictions. We also observed humans may be biased in some trials of deformability. But DPI-Net can handle these scenarios better, especially for trials in mass and deformability. Instead, models may fail in simple situations (due to lack of distance perception and intuitive physics) that are easier for humans to understand.
> > - Regarding 0.4 for hard trials and 0.7 on easy trials, it is defined by the mean human performance (L289-L293 in our main paper). We evaluate the "easy" trials by selecting stimuli from some of the same-initial-configuration pairs if one of the stimuli in the pair has mean human performance higher than 66.7%, and we also evaluate the "hard" trials where one of the paired stimuli has a mean human performance lower than 33.3%.  Therefore, hard trials are trials that humans have a high probability of failure, with a human performance of less than 50%.
> >
> > We will incorporate a detailed breakdown of the trials or properties in which humans exhibit better performance compared to baseline models, as well as cases where each model demonstrates superiority over human performance. We believe that this analysis will provide valuable insights into the capabilities and limitations of different methods and contribute to a more comprehensive physical scene understanding.

---

> > > ### Author Response · Authors · 2023-08-23
> > > **Rebuttal by Authors (Part 2/2)**
> > >
> > > **Q2. [Missing Baselines : The OCP task can be made conditionally independent of the visual information given the location, size, and tracks of the objects involved, detection of the contact event and the latent mechanical properties (even if specified qualitatively or comparatively in a paired setting). It seems that the difficulty of inferring the latter (observability?) and using them for reasoning (including inherent uncertainties involved in extrapolating on motion, force and dynamics) are what makes this task challenging both for humans and the machines. I’d have preferred baseline models and human experiments that decoupled these two aspects of the problem. (I’m happy to elaborate but want to keep the review short).]** \
> > > We appreciate your thoughtful suggestions! However, we'd like first to emphasize that this separation might not be feasible for humans. Humans are better at understanding visual information than symbolic states. Moreover, human reasoning inherently involves a holistic understanding of the world, where aspects such as visual cues and mechanical reasoning interplay seamlessly.
> > >
> > > Regarding baseline models and the consideration of different architectures, whether to explicitly include the latent mechanical properties or visual cues in a model's architecture may have some impacts on its performance. For example, DPI-Net might have an explicit advantage due to its design for accessing the object states rather than visual information, other models may not inherently possess this capability. However, our intention in constructing the Physion++ dataset was to create a holistic and generalized benchmark that encapsulates the real-world complexities of mechanical reasoning. This comprehensive approach inherently encompasses the challenges of inferring latent mechanical properties from visual information, and we are trying to ensure a fair comparison between different models and humans.
> > >
> > > We have also conducted an additional explicit property estimation experiment. Specifically, we selected four representative models (DeiT-mlp, ResNet-mlp, ALOE, and SlotFormer) for the experiment. A parallel layer of MLP is added at the end of the model (two-branch multitask: OCP and explicit property estimation) during the readout fitting process. With explicit property estimation as an auxiliary task, we report the performance comparison with our original setting as follows.
> > >
> > > | Method | DeiT-mlp | ResNet-mlp | ALOE | SlotFormer |
> > > |------|----------|------------|------|------------|
> > > | implicit (DCP only) | 55.4     | 55.1       | 53.4 | 56.7       |
> > > | explicit as auxiliary | 54.2     | 55.7       | 54.7 | 56.7       |
> > >
> > > From the results, we see explicit property estimation shows a minor effect. It brings slight improvement in some models, while decreases in others. Thank you again for your suggestions, and we believe that maintaining the holistic integrity and fairness of our benchmark aligns with our broader objective of advancing general physical reasoning capabilities.
> > >
> > >
> > > **Q3. [Human Explanations, Theories at Play: I also recommend that the authors think about collecting explanations from the human subjects which will allow them to gather data on what theories humans bring into play by themselves, and see if suggestions/ prompting/ guidance to use the right theory helps improve performance. This can be very helpful in rationalizing expectations from and the designing of such machine systems.]** \
> > > Good suggestion! We agree that human feedback and explanations are very helpful for a thorough understanding in cognitive science. Our current work focuses more on the benchmark itself and the baseline evaluation. We will consider involving more human feedback as our future work!
> > >
> > >
> > > **Q4. [Clarity: In Figure 4, bottom row paper (IV, V, VI): Kindly explain the human performance correlation with average human performance.]** \
> > > Thanks. We use the split-half Pearson's correlation for Figure 4 (IV, V, VI), as in Line 283 in our main paper. The human performance correlation means that different human participants' predictions on the same trial also vary. We will make it clearer in the revision.
> > >
> > >
> > > **Q5. [Reproducibility: Sharing the code to reproduce the results and analysis in the paper will go a long way in helping the community quickly and easily reproducing the work. Strongly urged.]** \
> > > We have uploaded our code to a [dropbox link](https://www.dropbox.com/scl/fi/nkz4jalxwtax89or0lunm/code_for_physion_v2.zip) and all the datasets to our [webpage](https://dingmyu.github.io/physion_v2/). We will further clean the code and release the trained baseline models together to Github soon.
> > >
> > >
> > > \
> > > Thanks again for your time and effort! For any other questions, please feel free to let us know during the rebuttal window.
> > >
> > > Best, \
> > > Authors

---

> ### Author Response · Authors · 2023-08-29
> **Looking forward to more discussions and your post-rebuttal rating!**
>
> Dear reviewer,
>
> We deeply appreciate the time you dedicated to reviewing our work and the insightful feedback you provided. We have crafted detailed responses and updated our results based on your comments to address your concerns.
>
> As the rebuttal period is ending soon, we wonder if our response answers your questions and addresses your concerns. Please do not hesitate to let us know if there are any additional clarifications that we can offer to make the paper better. We are always ready to address your further concerns and improve our draft as much as we can before the end of the rebuttal period on the 29th. Thanks again for your time and very constructive feedback!
>
> Best, \
> Authors

---

### Official Review · Reviewer_MSm3 · 2023-07-28
**A novel benchmark for testing inference of physical properties.**

**Rating:** 7
**Confidence:** 3
**Clarity:** Yes.

**Strengths:**

The paper is well-motivated. The authors thoroughly tested SOTA models on their benchmark. They provide performance comparisons with human subjects, indicating the gap in reasoning between DNNs and humans.

**Additional Feedback:**

None.

**Correctness:**

In the discussion, the authors say that none of the models match human performance, but it seems like DPI-Net does even better, so that sentence is not quite correct?

**Documentation:**

Yes.

**Ethics:**

No.

**Limitations:**

Yes.

**Opportunities For Improvement:**

The number of trials (500/property) seems relatively small for NN training. It would also be interesting to see an ablation of the number of trials over performance to see if training data is a limiting factor in the current models' performance or if it is more architecture related.

Is it possible to also directly assess a model's estimate of a physical property explicitly rather than implicitly, maybe as auxiliary tasks/metrics?

Figure 4: I would be interested in a deeper analysis of, for example, the differences between humans and DPI-Net; while on some tasks, the accuracy seems similar, the correlation is low, indicating they do well in different domains? Would it be possible to highlight properties/trials in which either human does better than DPI-Net and vice versa?

**Relation To Prior Work:**

No. Table 1 has a column 'Diverse Phenomena'. This term is never referred to/explained in the main text. The same applies to few-shot reasoning. As of now, these are the two entries in the table discriminating physion++ from IntPhys. It would therefore be very valuable to explain these or refer to them in the main text, highlighting how physion++ is different.

**Summary And Contributions:**

The authors introduce a novel benchmark for testing a model's ability to infer individual objects' properties such as mass from videos of diverse scenes.

---

> ### Author Response · Authors · 2023-08-23
> **Rebuttal by Authors (Part 1/2)**
>
> Dear Reviewer MSm3,
>
> Thank you for the positive comments and insightful suggestions! Your insightful questions and valuable suggestions have been immensely helpful in enhancing our paper's quality.
>
> **Q1. [The number of trials (500/property) seems relatively small for NN training. It would also be interesting to see an ablation of the number of trials over performance to see if training data is a limiting factor in the current models' performance or if it is more architecture related.]** \
> Sorry for the confusion. The number of trials is 2000 per property and we have total 8000 trials for the training set, as we detailed in the supplementary material (line 101). "500/property" is a typo in our paper, we apologize for any confusion this may have caused. Also, we have released all the datasets on our [webpage](https://dingmyu.github.io/physion_v2/), including training/readout/test set and the human data. Please feel free to check it out.
>
> Following your suggestions, we also conducted additional ablative experiments with different numbers of trials. Due to time constraints, we selected four representative models (DeiT-mlp, ResNet-mlp, ALOE, and SlotFormer) for experiments. The models learn dynamics separately for each mechanical property with property inference, and we report the average performance of the four mechanical properties. The results are shown in the below table.
>
> | Number of training trials per property | DeiT-mlp | ResNet-mlp | ALOE | SlotFormer |
> |------|----------|------------|------|------------|
> | 200  | 49.0     | 50.0       | 50.5 | 51.6       |
> | 500  | 53.6     | 51.6       | 52.1 | 54.2       |
> | 1000 | 54.7     | 54.7       | 54.2 | 56.3       |
> | 2000 | 55.4     | 55.1       | 53.4 | 56.7       |
>
> We can see that:
> - When the data size is small (200), the performance of all models is close to random guessing.
> - As the number of training data increases, the accuracy of the model improves and gradually saturates between 1000 to 2000. Therefore, we believe that 2000 training trials for each property are reasonable, and the performance difference in this case is more architecture related.
> - For models DeiT-mlp and ResNet-mlp, it seems like more data could lead to further performance growth. In the future, we will continue to enrich our data to validate this and evaluate more models.
>
>
> **Q2. [Is it possible to also directly assess a model's estimate of a physical property explicitly rather than implicitly, maybe as auxiliary tasks/metrics?]** \
> Thanks for the suggestion! We first state the reason that we use the object contact prediction (OCP) task for implicit property inference. Considering that different methods have different model designs, the architecture and learning difficulty of adding explicit property estimation vary, however, by using OCP, we put all methods on the same page. We will add the clarification in our main paper.
>
> We also try our best to do the explicit property estimation as you suggested. We selected four representative models (DeiT-mlp, ResNet-mlp, ALOE, and SlotFormer) for the experiment. A parallel layer of MLP is added at the end of the model (two-branch multitask: OCP and explicit property estimation) during the readout fitting process. With explicit property estimation as an auxiliary task, we report the performance comparison with our original setting as follows.
>
> | Method | DeiT-mlp | ResNet-mlp | ALOE | SlotFormer |
> |------|----------|------------|------|------------|
> | implicit (OCP only) | 55.4     | 55.1       | 53.4 | 56.7       |
> | explicit as auxiliary | 54.2     | 55.7       | 54.7 | 56.7       |
>
> From the results, we see explicit property estimation shows a minor effect. It brings slight improvement in some models, while decreases in others.

---

> > ### Author Response · Authors · 2023-08-23
> > **Rebuttal by Authors (Part 2/2)**
> >
> > **Q3. [Figure 4: I would be interested in a deeper analysis of, for example, the differences between humans and DPI-Net; while on some tasks, the accuracy seems similar, the correlation is low, indicating they do well in different domains? Would it be possible to highlight properties/trials in which either human does better than DPI-Net and vice versa?]** \
> > Good suggestion! We agree that a comprehensive analysis of the variations between human and DPI-Net performance is essential. Our current findings suggest that while the accuracy levels may be comparable, the correlation between their performances is low, indicating their proficiency in distinct domains. We have conducted a more thorough analysis of the data for highlighting instances where either humans or DPI-Net outperform the other.
> >
> > The representative videos are uploaded to our [webpage](https://dingmyu.github.io/physion_v2/). We can see that humans tend to have trouble dealing with unusual camera angles, contrasting physical properties (e.g., two objects with one very heavy and one very light, or very large elasticity), and long-term predictions. But DPI-Net can handle these scenarios better, especially for trials in mass and deformability. Instead models may fail in simple situations (due to lack of distance perception and intuitive physics) that are easier for humans to understand.
> >
> > In our revised manuscript, we will incorporate a detailed breakdown of the trials or properties in which humans exhibit better performance compared to baseline models, as well as cases where each model demonstrates superiority over human performance. We believe that this analysis will provide valuable insights into the capabilities and limitations of different methods and contribute to a more comprehensive physical scene understanding.
> >
> >
> > **Q4. [In the discussion, the authors say that none of the models match human performance, but it seems like DPI-Net does even better, so that sentence is not quite correct?]** \
> > Thanks for pointing this out, we'll rephrase the sentence. Their overall performances are comparable.
> >
> >
> > **Q5. [Table 1 has a column 'Diverse Phenomena'. This term is never referred to/explained in the main text. The same applies to few-shot reasoning. As of now, these are the two entries in the table discriminating physion++ from IntPhys. It would therefore be very valuable to explain these or refer to them in the main text, highlighting how physion++ is different.]** \
> > We apologize for any confusion this may have caused and will make it clearer in our revision. Regarding 'Diverse Phenomena', we refer to whether the dataset covers diverse scenarios. As shown in Figure 1, our dataset has 9 scenarios, including rigid bodies, fluids, soft bodies, objects of various shapes, and various physical properties. Previous physical reasoning datasets such as Comphy and CLEVRER only have single or two scenarios with few object shapes.
> >
> > As for 'Few-shot Reasoning', we mean that our dataset allows physical properties to be judged from a few reference video frames (inference phase in Figure 2). Completing the OCP task based on few-shot reference videos can also be understood as property inference. We'll make it clearer.
> >
> > \
> > Thanks again for your time and effort! For any other questions, please feel free to let us know during the rebuttal window.
> >
> > Best, \
> > Authors

---

> > > ### Author Response · Authors · 2023-08-29
> > > **Looking forward to more discussions and your post-rebuttal rating!**
> > >
> > > Dear Reviewer,
> > >
> > > We deeply appreciate the time you dedicated to reviewing our work and the insightful feedback you provided. We have crafted detailed responses and updated our results based on your comments to address your concerns.
> > >
> > > As the rebuttal period is ending soon, we wonder if our response answers your questions and addresses your concerns. Please do not hesitate to let us know if there are any additional clarifications that we can offer to make the paper better. We are always ready to address your further concerns and improve our draft as much as we can before the end of the rebuttal period on the 29th. Thanks again for your time and very constructive feedback!
> > >
> > > Best, \
> > > Authors

---

### Author Response · Authors · 2023-08-23
**Author Rebuttal by Authors: General Response**

Dear ACs and Reviewers,

We sincerely appreciate all reviewers’ time and efforts in reviewing our paper. We are glad to find that reviewers generally recognized our contributions:
- **Idea.** well-motivated [MSm3], Physion++ is a step in deducing about mechanical properties of objects from their dynamics, sensed and perceived visually, and using them to make physical predictions [zV2y], spurring more research in this important direction [zV2y], intuitive and interesting [vxoX], can be impactful [vxoX]
- **Experiment.** thoroughly tested SOTA models on their benchmark [WLS8, vxoX, MSm3], indicating the gap in reasoning between DNNs and humans [MSm3], Human evaluations are provided [WLS8, vxoX, MSm3]
- **Dataset.** should be reproducible given the information in the paper, supplementary and the website [zV2y], dataset and the benchmark to be useful to the research community [zV2y], the methodology is ok [zV2y]
- **Writing.** largely written well [zV2y], very good [vxoX]

And we thank all reviewers for their insightful and constructive suggestions, which help a lot in further improving our paper. In addition to the pointwise responses below, we summarize supporting experiments added in the rebuttal according to reviewers’ suggestions.

**Summary of new experiments or analysis**.
- Ablation on the number of training trials [MSm3]
- Ablation on explicit property estimation [MSm3,zV2y]
- Thorough analysis for highlighting trials where either humans or models outperform each other, with demo videos in our [webpage](https://dingmyu.github.io/physion_v2/) [MSm3]
- Release the initial code at the [dropbox link](https://www.dropbox.com/scl/fi/nkz4jalxwtax89or0lunm/code_for_physion_v2.zip) [zV2y]
- Explanations for relatively low human and model performance [zV2y, vxoX, WLS8]
- Experiments for humans predicting the state of the same scene [WLS8]

We hope these new additions help address reviewer concerns and better position our work. We thank the reviewers' time and feedback in improving the quality of our work. Please let us know if any clarification or additional experiments would further strengthen the paper. We would be happy to incorporate all these suggestions in the final version. Thank you again for your time and efforts!

Best, \
Authors

---

### Decision · Program_Chairs · 2023-09-22

**Decision:**

Accept (Poster)

**Comment:**

This paper proposes Physion++, a dataset and benchmark for evaluating a model's ability to predict latent physical properties such as mass and bounciness from sequence of visual input.  The dataset consists of various scenarios of interacting objects simulated using ThreeDWorld.  Experiments compare models that are trained to predict whether two object will collide (after an initial video of the two objects interacting).

Overall, reviewers are positive on the work (all vote for acceptance).  Reviewers found the paper to be well-motivated and appreciated the design of the benchmark.  During the author response and discussion period, the authors provided several different ablations, provided additional explanations and initial code, helping to convince the reviewers of the value of the work.

The AC agrees the work is of interest and step toward evaluating models that can infer physical properties and recommends acceptance.

The AC also recommends that the authors incorporates the reviewer feedbacks, extra experiments, and revise the paper to clarify unclear points.  There are sufficient questions from the reviewers that the paper does need to be revised.  Short summary is given below (see discussion for more details):
- Add ablations and experiments included in rebuttal
- Add discussion of human performance vs model performance
- Fix typos and improve clarity of writing (see MSm3-Q1,Q4,Q5, zV2y-Q4, WLS8-Q1)